# Cardiovascular Risk in Patients with Dyslipidemia and Their Degree of Control as Perceived by Primary Care Physicians in a Survey—TERESA-Opinion Study

**DOI:** 10.3390/ijerph20032388

**Published:** 2023-01-29

**Authors:** Vicente Pallarés-Carratalá, Vivencio Barrios, David Fierro-González, Jose Polo-García, Sergio Cinza-Sanjurjo

**Affiliations:** 1Health Surveillance Unit, Unión de Mutuas, 12004 Castellón de la Plana, Spain; 2Department of Medicine, Universitat Jaume I, 12071 Castellón de la Plana, Spain; 3Cardiology Department, H Ramón y Cajal, 28034 Madrid, Spain; 4Department of Medicine, Alcala University, 28801 Madrid, Spain; 5Armunia Health Centre, 24009 León, Spain; 6Casar de Cáceres Health Centre, 10190 Cáceres, Spain; 7Milladoiro Health Centre, 15895 Santiago de Compostela, Spain; 8Instituto de Investigación de Santiago de Compostela (IDIS), 15706 Santiago de Compostela, Spain; 9Centro de Investigación Biomédica en Red-Enfermedades Cardiovasculares (CIBER-CV), 28029 Madrid, Spain

**Keywords:** cardiovascular risk, dyslipidemia, statins, side-effects

## Abstract

Objective: The aim of this study was to evaluate, through a survey, the opinion of primary care (PC) physicians on the magnitude of dyslipidemia and its degree of control in their clinical practice. Materials and methods: An ecological study was carried out, in which the physicians were invited to participate by means of an online letter. Data were collected at a single timepoint and were based only on the experience, knowledge, and routine clinical practice of the participating physician. Results: A total of 300 physicians answered the questionnaire and estimated the prevalence of dyslipidemia between 2% and 80%. They estimated that 23.5% of their patients were high-risk, 18.2% were very high-risk, and 14.4% had recurrent events in the last 2 years. The PC physicians considered that 61.5% of their patients achieved the targets set. The participants fixed the presence of side-effects to statins at 14%. The statin that was considered safest with regard to side-effects was rosuvastatin (69%). Conclusions: PC physicians in Spain perceive that the CVR of their patients is high. This, together with the overestimation of the degree of control of LDL-C, could justify the inertia in the treatment of lipids. Moreover, they perceive that one-sixth of the patients treated with statins have side-effects.

## 1. Introduction

According to the World Health Organization (WHO), cardiovascular diseases (CVD) are the leading cause of mortality in the world, representing about 30% of annual deaths [1]. In CVD prevention, it is necessary to correctly identify the patients’ cardiovascular risk (CVR) using the SCORE-2 (Systematic Coronary Risk Evaluation) [2] scales, which estimate the 10 year risk of cardiovascular death, as well as to properly control cardiovascular risk factors (CVRF) on the basis of that risk [3].

The prevalence of dyslipidemia at the community level is 18.6% [4], increasing to 50.3% in the clinical population [5]. In the last 30 years, a change has been observed in this prevalence, with a slight decrease in high-income countries and an important increase in low-income countries [6]. This global perspective can potentially guide countries in the development of their own risk assessment models and in the elaboration of recommendations in their own guidelines according to local requirements [6]. We can also observe this association between social factors and prevalence of dyslipidemia when we analyze the data in each country, with a higher prevalence among women, probably because of a higher prevalence of obesity, and among populations with low income and low educational level [4]. On the other hand, in developed countries, the prevalence of CVD increases with age, and it can, for example, increase the fragility of patients and worsen their prognosis [7,8,9]. 

In CVR prevention strategies, a most influential factor is the control of low-density lipoprotein cholesterol (LDL-C) levels, which is based on treatment with statins. The use of statins has been shown to effectively reduce LDL-C and, therefore, CVR, especially high-intensity statins such as atorvastatin (ATV) and rosuvastatin (RSV) [10]. Improving the control of dyslipidemia and the use of statins are the most important strategies in patients with CVD [3], because statins can not only reduce LDL-C but also have a pleiotropic effect on reducing oxidative stress, which has a role in the physiopathology of atherosclerosis, as well as in restenosis in patients with coronary disease [11].

In clinical practice, different studies have shown that a high proportion of patients do not achieve the targets indicated [12] in the clinical practice guidelines [3,13]. In Spain, the degree of control of dyslipidemia ranges from 13% [14] to 26% [15]. The targets set by the clinical practice guidelines [3] are hard to reach [16], since there are other reasons that explain this low degree of control of LDL-C, such as (diagnostic or therapeutic) inertia on the part of the physicians [17,18,19] and lack of adherence to treatment on the part of the patients [20]. In light of this situation of poor degree of control, there have been important developments and ongoing research in the last three decades that will expand the available treatment options and will enable further cardiovascular risk reduction [21]. Thus, clinicians have the obligation to change their practices in order to improve prognosis in their patients, especially with CVD.

From our point of view, one of the causes of diagnostic–therapeutic inertia is the physicians’ perception of the risk associated with dyslipidemia; the patient’s CVR is underestimated, and the degree of control of LDL-C levels reached is overestimated [22]. This is why we propose this study, with the objective of assessing, through a survey among primary care (PC) physicians, their opinion on the magnitude of dyslipidemia and on the degree of control among patients in their clinical practice.

## 2. Materials and Methods

### 2.1. Design of the Study

An ecological study was carried out in which information was collected on the opinions and assessments of the participating physicians. Data were collected at a single timepoint and were based only on the experience, knowledge, and routine clinical practice of the participating physician. We did not collect any data from patients.

We selected the physicians randomly and in proportion to the number of physicians in the Autonomous Communities to get a representative number of answers in each one. 

### 2.2. Sample

PC physicians from across the country who are currently working in the National Health System were invited to participate (according to information from the Ministry of Health, the total number of PC physicians in Spain is 36,075). The invitation was sent online through a presentation of the study which provided a link to the survey. No exclusion criteria were considered, except refusal to participate. In total, 302 PC physicians (0.8%) answered the survey, but two of them did so incompletely. Therefore, we included 300 answers (0.8% of the total of PC physicians in Spain) in the final analyses.

### 2.3. Variables

The source of information was the physicians themselves, who answered the questions in the questionnaire according to their experience, knowledge, and routine clinical practice. The variables recorded were those corresponding to the survey presented in the Appendix A.

### 2.4. Ethical Aspects

Since the data collection was retrospective and the data were pooled, there was no interference with the physician’s prescribing habits. The data derived from the study were epidemiological aggregated data which in no case would come from the patients’ medical history.

This study was approved on 4 August 2021 by the CREC of Hospital Clínico San Carlos with the code 21/558-E.

### 2.5. Statistical Analysis

For the quantitative variables, the Shapiro–Wilk test was used to check the data’s fit to a normal distribution. If the variable showed a normal distribution, it was described using the arithmetic mean and the standard deviation (SD); otherwise, the median and the interquartile range [Q1, Q3] were used. 

In the descriptive analysis of the qualitative variables, results are presented as percentages. In questions where respondents were asked to order by preference, it was checked that participants had ordered all the items; thus, if at least one item had no reply, this question was excluded from the preference analysis. For each characteristic assessed as a preference, results are presented as the percentage of investigators with a reply in that characteristic, ranging from 1 “highest preference” to X “lowest preference”. Thus, a higher score assigned was reflective of a lower preference for the item assessed. 

The statistical analysis was performed with the statistical package IBM SPSS version 28.

## 3. Results

### 3.1. Physicians Who Responded to the Survey

The analysis included the responses of 300 PC physicians with a mean age of 53.2 (10.0) years and representing all the Spanish autonomous communities and autonomous cities (Table 1).

The majority of physicians worked in health centers (88.0%), whereas the remainder worked in rural practices (12.0%). Overall, 61% of these centers were teaching centers: 26.3% of physicians were attending physicians, and 29.7% received medical students. Most of the physicians had more than 1500 patients assigned, and 35.3% had between 1000 and 1499 patients. The remainder (9%) had <1000 patients assigned.

### 3.2. Magnitude of Dyslipidemia

The respondent physicians estimated that between 2.0% and 80% of the last 10 patients seen in their practices had dyslipidemia, with a mean of 36.9%, of which a majority (60.7%) were between 40 and 80 years old.

The most common therapeutic strategy in patients younger than 40 was moderate-intensity statins (33.4%), followed by high-intensity statins (23.1%); in patients aged 41–80, these proportions were 27.8% and 27.1%, respectively. However, in patients older than 80, the most widely used strategy was moderate-intensity statins (35.6%), which were used far more than high-intensity statins (19.8%), low-intensity statins (19.7%), or the combination of moderate-intensity statins and ezetimibe (18.7%). The combination with ezetimibe was most common in patients aged 40 to 80 (Figure 1). Only 1.4% of physicians recalled patients with PCSK9 inhibitors. 

### 3.3. Comorbidities in Patients with Dyslipidemia

The association with other comorbidities is shown in Figure 2, where it can be observed that T2DM and obesity were most frequent, followed by HTN and coronary heart disease. In this context, physicians were also requested to classify their patients according to the presence of CVR-modifying diseases. 

The physicians who responded to the survey estimated that 21.7% of their patients were low-risk, 28.5% were intermediate-risk, and 23.5% were high-risk. The higher-risk patients were divided into 18.2% as very high risk and 14.4% as patients with recurring events in the last 2 years.

With regard to the most used therapeutic strategies for patients with dyslipidemia, the most common was RSV 10 mg, with the most preferential score in the majority of physicians, followed by RSV 20 mg and ATV 20 mg. Among the fixed combinations marketed in Spain, the most widely used was RSV/ezetimibe (36.0%), followed by ATV/ezetimibe (26.6%). The least used combinations were statin/fibrate (15.7%) and simvastatin/ezetimibe (14.1%). 

The highest-intensity statins and their combination with ezetimibe were most used in patients with CVD and with T2DM, whereas lifestyle habits and lower-intensity statins were most used in patients without comorbidities (Figure 3).

### 3.4. Opinion on the Degree of Control of Patients

The respondents considered median values of 60 [50, 70] as acceptable for the control of LDL-C in patients with recurring event, and values of 60 [55, 70] in very high-risk patients. They considered higher values of LDL-C acceptable in high-risk patients (70 [70, 100]), moderate-risk patients (100 [99, 100]), and low-risk patients (116 [100, 130]).

The PC physicians thought that 61.5% of their patients achieved the targets set in the clinical practice guidelines. Figure 4 shows the degree of control estimated by the physicians by interest group according to risk.

### 3.5. Safety of Statins

The participants estimated that the prevalence of side-effects to statins was 14% of cases, with a median of 10% [5%, 20%]. When a patient experienced side-effects to statins, 52.7% of physicians made modifications to the statin, with a median of 50% [10%, 96%]. Among these modifications, the most widely used strategy was to switch to another statin (40% [20%, 65%]) and to reduce the dose of statin (30% [10%, 50%]). In 20% [10%, 50%] of cases, the statin was withdrawn. Table 2 shows the strategies used when side-effects occurred.

The statin considered safest with regard to side-effects was RSV (69%), followed by pitavastatin (14%). Only 8% of physicians considered ATV as the safest statin.

## 4. Discussion

The results of our survey, conducted among 300 PC physicians, showed a widely varying perception of the prevalence of dyslipidemia, which ranged from 2.0% to 80%. Their perception regarding CVR is that the majority of patients belonged to the high- or very high-risk groups, probably because the most widely perceived comorbidity was T2DM with or without coronary heart disease. However, the most common strategy was moderate-intensity statins, whereas the highest-intensity statins were used mainly in patients aged 40 to 80. In this clinical and therapeutical context, the PC physicians estimated that almost two-thirds of the patients with dyslipidemia achieved the therapeutic targets, and that one-sixth of patients treated with statins presented side-effects. In this case, the most frequent strategy was to reduce the dose or switch to another statin. They considered RSV as the safest statin, which was also the statin most widely used by the respondents.

Although many surveys have been carried out among physicians and patients on the estimation of cardiovascular risk, we found, after an extensive literature review, that our study is the first to provide recent data on the opinion of PC physicians on CVR, comorbidities, degree of control, and drug treatment used and its safety. This allows us to check whether the responses are consistent with each other, and to explain certain prescription habits such as maintaining low-intensity statins in high-CVR patients who do not achieve targets or to withdraw the treatment with statins given the possibility of unproven side-effects. 

Our results show that the PC physicians perceived that more than half of their patients with dyslipidemia had a high or very high CVR, although they estimated a degree of control of 61.5% in these patients, and they chose moderate-intensity statins as the main therapeutic strategy. These findings are in line with those published recently by Barrios et al., whose survey showed that 60–64% of physicians considered that lipid levels were well controlled [23]. This perception of good control might justify the therapeutic inertia to maintain the treatment with moderate-intensity statins despite the high CVR of these patients [24,25], although the physicians considered that only one-third of their patients had high CVR.

In our study, the physicians identified T2DM and obesity as the comorbidities most frequently associated with dyslipidemia and, together with CVD, the conditions which most frequently modified their patients’ CVR. This response is consistent with epidemiological data published in the IBERICAN study, conducted among PC physicians in Spain [26], and with other surveys, such as Plana et al., where physicians prioritized DM and CVD, without mentioning obesity [27]. The responses provided seem not only to be in line with reality and with published data, but also to be sincere, since they acknowledge that the most widely used strategy is moderate-intensity statins, as in other developed countries [28]; this could justify the low degree of control of patients with dyslipidemia in our country [29]. We consider that, with the availability of new therapies, it is very important to know the degree of control of dyslipidemia bearing in mind the new targets, to reveal the actual situation and to take measures to improve the cardiovascular prognosis in our population [30].

The underestimation of CVR, caused by the fact that risk scales are not used, and that decision making is based on clinical instinct, brings about therapeutic inertia. This situation is worsened if, in addition, the patients themselves underestimate their own CVR, because this can compromise their adherence to lifestyle habits and drug treatments. In this regard, population-based surveys have shown that women tend to underestimate their own risk compared to men [31,32], and that they only perceive an increased risk with age, the accumulation of CVRF, or the presence of CVD [31]. 

A survey conducted by Chapman et al. showed that PC physicians acknowledged that they rarely evaluated the CVR of their patients, and that, when they used this estimation, it was to try to motivate the patient to change their lifestyle habits or to make some change in the drug treatment [33]. Moreover, most of the respondents considered that the lack of time prevented them from estimating the protocol CVR calculation, and that they drew on their “clinical instinct” to assess the risk. Therefore, 84% of them considered that, if the medical history reported the CVR from the information already recorded, this would avoid existing barriers [33]. In our opinion, this could be the strategy with the greatest impact on the reduction in the population’s CVR, because other strategies based on physicians’ training or awareness raising, such as the clinical trial by Van Steenskiste et al., have had little success and have not brought about changes in pharmacological strategies [34]. 

Our results also show that the most widely used therapeutic strategy was moderate-intensity statins, a lower step than that which would be expected for this risk according to the recommendations of the clinical practice guidelines [3], as already described in records such as Da Vinci, where only 20% of patients without CVD and 34% of patients with CVD received high-intensity statins [35]. A possible justification would be the lack of knowledge of the control targets on the part of physicians. However, in the survey, they claimed to have adequate knowledge of the therapeutic targets laid out in the guidelines, which is higher than the 53.3% of doctors surveyed by Rainer et al. [36]. Another reason that could explain this strategy is the underestimation of the patients’ CVR on the part of their physicians, as observed in other surveys [37], which, as stated above, cannot be accepted in view of our results. A third explanation is that the inertia might have been the result of an overestimation of the patients’ degree of control; therefore, there was no perceived need to intensify the treatment, whereas the reality is very different with LDL control figures lower than 25% [15,29,38].

Lastly, this lack of intensification could also be explained by the fear of side-effects. In our study, the presence of side-effects was found in 14% of cases. This figure is similar to that described in the SAMSON (Self-Assessment Method for Statin Side-Effects or Nocebo) study, which analyzed the presence of side-effects to statins in patients with chronic treatment. Howard et al. analyzed the presence of side effects in a 12 month crossover trial, in which all the patients received three therapeutic strategies for 4 months each: statins, placebo, and no treatment. The prevalence of side-effects observed during the months of treatment with statins (16.3; 95% CI: 13.0–19.6) was similar to the prevalence during placebo (15.4; 95% CI: 12.1–18.7, *p* = 0.388), and both were much higher than that of the period with no treatment (8.0; 95% CI: 4.7–11.3). Despite the proven safety of statins [39], results such as those from our survey or from the SAMSON study confirm that both patients and physicians have a higher perception than reality about the side-effects associated with statins [40]. This may be due to lack of adherence on the part of patients, and to the use of less aggressive lipid-lowering strategies on the part of physicians.

If side-effects occur, our respondents took measures (40% switch to another statin, and 30% reduce the dose of the same statin) similar to those observed in other surveys such as Gupta et al., where 38.1% of respondents switched to another statin and 30.7% reduced the dose [41]. 

From our point of view, in the case of Spain, it is very important to improve the treatment of dyslipidemia with statins and other drugs because the new treatments, such as PCSK9 inhibitors or, in the future, inclisiran and evinacumab, are restricted only to hospital treatment in patients with very high risk and that do not reach <100 mg/dL. For this reason, in practice, we have very few patients treated with PCSK9 inhibitors, which represent a marginal treatment compared to statins. 

Our work had, of course, some limitations. On the one hand, the typical limitation of every ecological study, i.e., the fact that all variables are aggregated, did not allow us to establish causal or time relationships. However, as stated before, we understand that the results provided are consistent with the published data and reasonably consistent with each other. On the other hand, since it was a voluntary survey, the physicians who were more sensitive to CVR were more likely to have responded to the survey, potentially showing a greater knowledge and therapeutic strategies more aligned with clinical practice guidelines. In any case, the design of the study does not nullify the principal objective proposed, because it gives a clear idea of the physicians’ opinion on the CVR of their patients, and of the clinical strategies adopted in their clinical practice. Lastly, the sample size was seemingly low, but it represented about 1% of the physicians in PC in Spain; moreover, they were selected randomly in each Autonomous Community. Therefore, we think our main objective was addressed because the results observed were also in line with those published in cohort studies and in other surveys.

## 5. Conclusions

In light of the results presented, we can conclude that the PC physicians in Spain correctly perceive that the CVR of their patients is high, although they make use of moderate-intensity therapeutic strategies, which, together with the overestimation of the degree of control of LDL-C, could justify the inertia in the treatment of lipids. On the other hand, they consider that one-sixth of the patients treated with statins have side-effects, and the most frequent management in these cases is reducing the dose of statins or switching to another statin.

## Figures and Tables

**Figure 1 ijerph-20-02388-f001:**
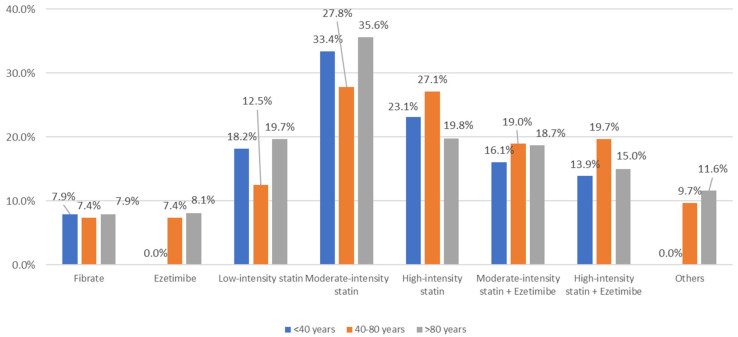
Pharmacological treatment considered by physicians for use in patients with dyslipidemia by age.

**Figure 2 ijerph-20-02388-f002:**
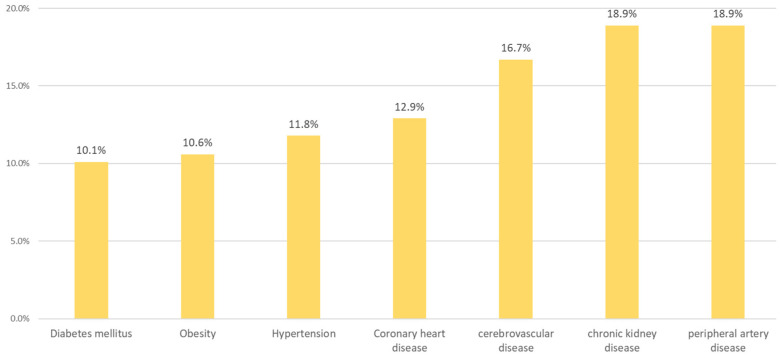
Comorbidities considered most frequent by the physicians.

**Figure 3 ijerph-20-02388-f003:**
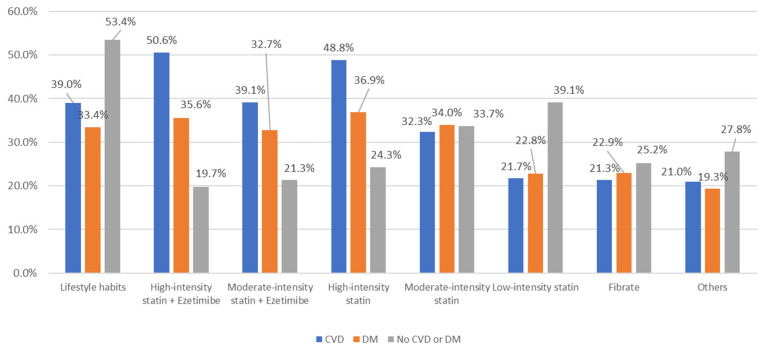
Treatment used by the physicians in patients with cardiovascular disease (CVD) and diabetes mellitus (DM), and in patients without any of these diseases. CVD: cardiovascular disease; DM: diabetes mellitus.

**Figure 4 ijerph-20-02388-f004:**
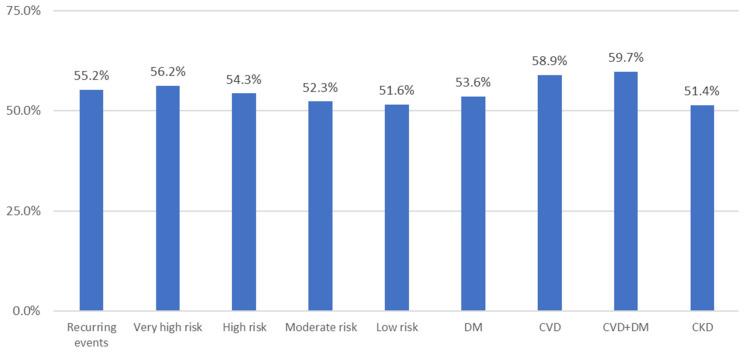
Rate of dyslipidemia control estimated by the physicians in different situations. DM: diabetes mellitus; CVD: cardiovascular disease; CKD: chronic kidney disease.

**Table 1 ijerph-20-02388-t001:** Distribution of physicians who responded to the survey by Autonomous Community.

		N	%
Autonomous Community	Andalusia	45	15.0
	Aragon	13	4.3
	Principality of Asturias	8	2.7
	Balearic Islands	6	2.0
	Canary Islands	16	5.3
	Cantabria	4	1.3
	Castilla y Leon	22	7.3
	Castilla La Mancha	16	5.3
	Catalonia	35	11.7
	Valencian Community	40	13.3
	Extremadura	10	3.3
	Galicia	23	7.7
	Community of Madrid	34	11.3
	Region of Murcia	11	3.7
	Foral Community of Navarra	3	1.0
	Basque Country	10	3.3
	La Rioja	1	0.3
	Autonomous City of Melilla	3	1.0
	Total	300	100.0

**Table 2 ijerph-20-02388-t002:** Strategies used when there were side-effects to statins.

%	Mean	SD	Median	Q1	Q3	Minimum	Maximum	N
Switch to nutraceutical	13.1	24.5	0.0	0.0	10.0	0.0	100.0	268
Switch to another statin	79.9	28.3	90.0	70.0	100.0	5.0	100.0	35
Switch to another statin or reduce the dose of the same statin	100.0	-	100.0	100.0	100.0	100.0	100.0	1
Switch to another drug	67.5	34.8	80.0	30.0	100.0	1.0	100.0	61
Reduce the dose of the same statin	65.0	52.2	90.0	5.0	-	5.0	100.0	3
Discontinue the statin	97.5	3.5	97.5	95.0	-	95.0	100.0	2
Others	52.3	42.4	40.0	7.5	95.0	5.0	100.0	13
Lifestyle habits	34.4	38.9	20.0	5.0	70.0	5.0	100.0	9
Lifestyle habits and switch to another drug	90.0	-	90.0	90.0	90.0	90.0	90.0	1
Others	93.3	5.8	90.0	90.0	-	90.0	100.0	3

## Data Availability

Data are contained within the article or Appendix A.

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
