# Peer review of "Cardiovascular Risk in Patients with Dyslipidemia and Their Degree of Control as Perceived by Primary Care Physicians in a Survey—TERESA-Opinion Study"

_ijerph, 2023, doi:10.3390/ijerph20032388_

Round 1

Reviewer 1 Report

Dear Authors,

thank you for submitting the paper and for the chance that I had to revise it. I have the following comments:

- the total number of responders is 300; which proportion of the all PC represent?

- does the PC know how many events did they avoid with statins? how many patients experienced CV complication such as stroke or acute myocardial infarction?

- at the beginning of the discussion section there is a default paragraph from the LaTeX template that has to be eliminated.

- the paper has a potential but should adhere for the reporting of the survey to the CROSS paper (10.1007/s11606-021-06737-1)

Author Response

On behalf of my co-authors, thank you for your recommendations regarding our article, "Cardiovascular risk in patients with dyslipidemia and their degree of control as perceived by primary care physicians” MANUSCRIPT: ijerph-2140087

We would also like to thank the referees for taking the time to read our manuscript and provide constructive feedback. We have addressed the comments, which are explained point by point, and the revisions to the manuscript appear as tracked changes.

We hope that these revisions are satisfactory to you and the referees.

Reviewers' comments:

Reviewer #1: MANUSCRIPT:  ijerph-2140087

- the total number of responders is 300; which proportion of the all PC represent?

In Spain there are approximately 35.000 PCP (https://www.sanidad.gob.es/estadEstudios/sanidadDatos/tablas/tabla13.htm). Our sample represents 1% of them.

- does the PC know how many events did they avoid with statins? how many patients experienced CV complication such as stroke or acute myocardial infarction?

No. Our work is a report on their idea about the degree of control of hypercholesterolemia, level of cardiovascular risk of their patients and how many patients they think have adverse events with statins. All of our results are the impressions of the PCP.

- at the beginning of the discussion section there is a default paragraph from the LaTeX template that has to be eliminated.

We are sorry for this mistake. We have deleted this paragraph.

- the paper has a potential but should adhere for the reporting of the survey to the CROSS paper (10.1007/s11606-021-06737-1)

We have reviewed the CROSS report and included it in the documentation of our manuscript.

Reviewer 2 Report

I received an original research article for review entitled "Cardiovascular risk in patients with dyslipidemia and their degree of control as perceived by primary care physicians", prepared by Vicente Pallarés Carratalá et al., which was submitted to the International Journal of Environmental Research and Public Health (IF=4.614). The article concerns a very important issue, which is cardiovascular risk. Cardiovascular disease is one of the most important healthcare problems worldwide, so research in this area is of crucial importance because it may lead to improve in the diagnosis and treatment of patients with cardiovascular disease. Primary care physicians play a huge role in the health care system because they are the ones who set the right track for the diagnostic and therapeutic process, which is why the subject matter taken up by the authors of the work should be considered important and up-to-date. The article is generally well prepared and represent some scientific value, but in my opinion significant modifications are necessary that may contribute to further improvement of the quality and attractiveness of the presented manuscript.

1)     I think that the introduction is too laconic and should be expanded. In my opinion, it is worth expanding the information on cardiovascular diseases. It is worth mentioning that the most important pathogenetic process that leads to the development of cardiovascular diseases is atherosclerosis. In the course of atherosclerosis, ischemic heart disease, cerebrovascular disease, and peripheral arterial disease develop. In the treatment of cardiovascular diseases in the course of atherosclerosis, angioplasty and stenting play an important role, but the process of restenosis is a phenomenon that significantly reduces the effectiveness of this treatment and may lead to the need for reintervention. LDL-C is the most important target in the pharmacological treatment of atherosclerotic cardiovascular disease, but statins also have a pleiotropic effect, which is associated with beneficial effects on the cardiovascular system, including stabilization of the atherosclerotic plaque and decrease in oxidative stress. It would be worth to mention that dyslipidemia is not only incorrect concentration of lipid parameters but also the presence of dysfunctional lipoproteins. For example nitrated lipoproteins has been recently discussed to take a part in the pathogenesis of cardiovascular disease. (10.3390/ijerph182211970; 10.3390/ijerph17249339; 10.1007/s11886-019-1175-z; 10.1001/jamanetworkopen.2021.27573.)

2)     The description of the statistical analysis methodology should be improved. Please describe how the correspondence between the empirical distribution of the quantitative variable and the normal distribution was examined.

3)     “Authors should discuss the results and how they can be interpreted from the perspective of previous studies and of the working hypotheses. The findings and their implications should be discussed in the broadest context possible. Future research directions may also be highlighted.” – This text should be removed from the discussion.

4)     The study is conducted in Spain and therefore refers to the realities of the healthcare system in this country. It is worth dedicating a separate part of the discussion to describing the availability of new lipid-lowering therapies in this country, such as PCSK9 inhibitors, inclisiran, and evinacumab. Describe whether these treatments are available in PC physicians. Such information is extremely important for readers from different countries and will significantly affect the interpretation of the results obtained.

5)     It should be corrected: “guidelines3”. (line 233)

6)     English should be revised and eventually corrected by a philologist.

Author Response

On behalf of my co-authors, thank you for your recommendations regarding our article, "Cardiovascular risk in patients with dyslipidemia and their degree of control as perceived by primary care physicians” MANUSCRIPT: ijerph-2140087

We would also like to thank the referees for taking the time to read our manuscript and provide constructive feedback. We have addressed the comments, which are explained point by point, and the revisions to the manuscript appear as tracked changes.

We hope that these revisions are satisfactory to you and the referees.

Reviewer #2: MANUSCRIPT:  ijerph-2140087

I received an original research article for review entitled "Cardiovascular risk in patients with dyslipidemia and their degree of control as perceived by primary care physicians", prepared by Vicente Pallarés Carratalá et al., which was submitted to the International Journal of Environmental Research and Public Health (IF=4.614). The article concerns a very important issue, which is cardiovascular risk. Cardiovascular disease is one of the most important healthcare problems worldwide, so research in this area is of crucial importance because it may lead to improve in the diagnosis and treatment of patients with cardiovascular disease. Primary care physicians play a huge role in the health care system because they are the ones who set the right track for the diagnostic and therapeutic process, which is why the subject matter taken up by the authors of the work should be considered important and up-to-date. The article is generally well prepared and represent some scientific value, but in my opinion significant modifications are necessary that may contribute to further improvement of the quality and attractiveness of the presented manuscript.

First of all, we would like to thank the reviewer’s comments on our paper. These comments were very useful for improving the clarity and quality of our work.

1) I think that the introduction is too laconic and should be expanded. In my opinion, it is worth expanding the information on cardiovascular diseases. It is worth mentioning that the most important pathogenetic process that leads to the development of cardiovascular diseases is atherosclerosis. In the course of atherosclerosis, ischemic heart disease, cerebrovascular disease, and peripheral arterial disease develop. In the treatment of cardiovascular diseases in the course of atherosclerosis, angioplasty and stenting play an important role, but the process of restenosis is a phenomenon that significantly reduces the effectiveness of this treatment and may lead to the need for reintervention. LDL-C is the most important target in the pharmacological treatment of atherosclerotic cardiovascular disease, but statins also have a pleiotropic effect, which is associated with beneficial effects on the cardiovascular system, including stabilization of the atherosclerotic plaque and decrease in oxidative stress. It would be worth to mention that dyslipidemia is not only incorrect concentration of lipid parameters but also the presence of dysfunctional lipoproteins. For example, nitrated lipoproteins has been recently discussed to take a part in the pathogenesis of cardiovascular disease. (10.3390/ijerph182211970; 10.3390/ijerph17249339; 10.1007/s11886-019-1175-z; 10.1001/jamanetworkopen.2021.27573.)

We have completed the introduction with information about the prevalence, the degree of control, the physiopathology of atherosclerosis, the prognosis in patients with CVD and the new treatments. Also, we completed the number of bibliographic references.

2) The description of the statistical analysis methodology should be improved. Please describe how the correspondence between the empirical distribution of the quantitative variable and the normal distribution was examined.

We have reviewed and completed the methodology section to improve understanding. We have also included a better explanation about the statistical analyses that we did.

In addition, we have changed the format of figure 2 for a better understanding of its content. The methodology is explained in section 2.5, 2nd paragraph.

3) “Authors should discuss the results and how they can be interpreted from the perspective of previous studies and of the working hypotheses. The findings and their implications should be discussed in the broadest context possible. Future research directions may also be highlighted.” – This text should be removed from the discussion.

We are sorry for this mistake. We have deleted this paragraph.

4) The study is conducted in Spain and therefore refers to the realities of the healthcare system in this country. It is worth dedicating a separate part of the discussion to describing the availability of new lipid-lowering therapies in this country, such as PCSK9 inhibitors, inclisiran, and evinacumab. Describe whether these treatments are available in PC physicians. Such information is extremely important for readers from different countries and will significantly affect the interpretation of the results obtained.

We agree with this comment, and we have included a paragraph at the end of the discussion section to explain the situation in Spain regarding the new strategies on dyslipidaemia.

5) It should be corrected: “guidelines3”. (line 233)

Thanks for your observation. It is the reference number. We change the format.

6) English should be revised and eventually corrected by a philologist.

Our philologist is a specialist in scientific English and reviewed the first version, and also these comments and the new version that we send with theses answers to the editor and reviewers.

Round 2

Reviewer 2 Report

I believe that the Authors properly addressed the suggestions expressed in the review, and as a result, the quality of the presented manuscript increased significantly. I believe that this article in its final version is well prepared and has significant scientific and cognitive value. I recommend the article for publication in its current form.